# A hypothesis on the capacity of plant odorant-binding proteins to bind volatile isoprenoids based on in silico evidences

Deborah Giordano[1], Angelo Facchiano[1], Sabato D'Auria[1,2]*, Francesco Loreto[3,4]*

[1]Institute of Food Science, CNR, Avellino, Italy; [2]Department of Biology, Agriculture and Food Sciences, CNR, Rome, Italy; [3]Department of Biology, University of Naples Federico II, Naples, Italy; [4]Institute for Sustainable Plant Protection, CNR, Florence, Italy

**Abstract** Volatile organic compounds (VOCs) from 'emitting' plants inform the 'receiving' (listening) plants of impending stresses or simply of their presence. However, the receptors that allow receivers to detect the volatile cue are elusive. Most likely, plants (as animals) have odorant-binding proteins (OBPs), and in fact, a few OBPs are known to bind 'stress-induced' plant VOCs. We investigated whether these and other putative OBPs may bind volatile constitutive and stress-induced isoprenoids, the most emitted plant VOCs, with well-established roles in plant communication and defense. Molecular docking simulation experiments suggest that structural features of a few plant proteins screened in databases could allow VOC binding. In particular, our results show that monoterpenes may bind the same plant proteins that were described to bind other stress-induced VOCs, while the constitutive hemiterpene isoprene is unlikely to bind any investigated putative OBP and may not have an info-chemical role. We conclude that, as for animal, there may be plant OBPs that bind multiple VOCs. Plant OBPs may play an important role in allowing plants to eavesdrop messages by neighboring plants, triggering defensive responses and communication with other organisms.

*For correspondence:
sabato.dauria@cnr.it (SD'A);
francesco.loreto@unina.it (FL)

## Introduction

Plants synthesize a variety of volatile organic compounds (VOCs) that are important for reproduction and defense and in general to communicate with other organisms (*Ninkovic et al., 2021*). Insects and generalist herbivores, or carnivore insects that are also attracted by the volatile 'cry for help' released by plants upon herbivore attacks, are all able to sense plant volatiles (*Dicke and Loreto, 2010*).

Whether volatiles are also important in plant–plant communication is a more fascinating, yet controversial, issue (*Vickers et al., 2009*). Growing reports show that VOCs influence plant–plant relationships (*Baldwin et al., 2002*; *Erb, 2019*; *Ninkovic et al., 2019*). VOCs elicited in 'emitting' plants by abiotic or biotic stresses prime defensive responses in non-elicited 'receiving' plants (*Zuo et al., 2019*; *Frank et al., 2021*). However, no study has so far looked for the primary events in such elusive plant–plant interaction, i.e., the receptors by which plants may sense VOCs emitted from neighboring plants are largely unknown.

Recently, it has been proposed that the passage of VOCs across the plasma membrane relies on their active transport. In particular, the presence of an ABC carrier protein involved in active transport into plant cells has been hypothesized (*Adebesin et al., 2017*). Plant VOC receptors may belong to a similar category of transporters. An alternative explanation is that plants use odorant-binding proteins (OBPs) as protein carriers, alike animals. Indeed, there have been at least three cases in which the presence of OBPs was postulated in plants. These are as follows: (1) the COI1

assembly with a jasmonate zinc finger inflorescence meristem (ZIM)-domain (JAZ) protein family (COI1–JAZ), a high-affinity receptor protein for methyl-jasmonate (MeJa), and the volatile moiety of jasmonic acid (JA) (*Sheard et al., 2010*). After JAZ degradation, transcription factors are released, which activate downstream genes and the defensive metabolites in plants challenged by abiotic and biotic stresses (*Cheong and Choi, 2003*). (2) The salicylic acid (SA)-binding protein 2 (SABP2), an esterase of the a/b-fold hydrolase superfamily, that binds SA with high affinity and then converts the biologically inactive methyl ester of SA (MeSA) to active SA-inducing systemic acquired resistance in plants challenged by stresses (*Park et al., 2007*). (3) The TOPLESS-like protein (TPL) that specifically binds β-caryophyllene, a stress-induced sesquiterpene and a volatile signal for herbivores and carnivores in multitrophic interactions. TPL and TPL-related proteins are transcriptional co-repressors (also toward JA-mediated signaling). Interestingly, only the capacity to bind β-caryophyllene was tested with emitting and receiving (eavesdropping) plants (*Nagashima et al., 2019*).

These three cases need confirmation, and all other plant VOCs (at least 1700 known so far; *Dicke and Loreto, 2010*) wait for identification of receptors (if present). We report here an in silico study based on current knowledge of plant protein structure, especially aiming at identifying best candidates as plant OBPs for plant VOCs whose receptors are still unknown. We focused on the volatile isoprenoids (isoprene and monoterpenes) produced by the methyl erythritol phosphate (MEP) chloroplast pathway and representing the largest component of plant VOC emissions in the atmosphere (*Loreto and Schnitzler, 2010*).

## Results and discussion

We looked at plant OBPs following a two-level structural approach (identification of candidate plant OBPs, and validation by in silico molecular simulation experiments of OBP capacity), as detailed in the Materials and methods section. For the first level, we added to the three known plant OBPs those potential OBPs resulting from plant protein sequence databases (approach a) and from comparison for sequence similarities with known animal OBPs (approach b, see Materials and methods for details).

Approach (a) yielded five complete or partial protein sequences, three from *Anthurium amnicola* (named OBP56d, OBP A10_1, and OBP A10_2), one from *Nymphaea thermarum* (named putative OBP), and one from *Pyrus x bretschneideri* (named OBP-70 like) (*Supplementary file 1*). However, when these sequences were further screened for similarities with annotated plant proteins in the sequence databases, statistically relevant similarity and coverage was only found between the putative OBP from *N. thermarum* and the Flowering locus T (FT) and T1 proteins, and the heading date 3A and 3B. Comparison with animal OBPs, yielded similarities between (1) OBPs from insects and plant OBP56d and OBP-70 like, (2) chemosensor proteins and plant OBP A10_1, and (3) phosphatidylethanolamine-binding proteins and the putative OBP from *N. thermarum* (*Supplementary file 2*).

Approach (b) was based on comparison between 432 OBP protein sequences from different animal sources with plant protein sequences. Only five animal protein families share sequence similarity with plant proteins, as summarized in *Table 1*. Sequences were considered similar when showing BLAST E-values close to 0, and sequence identity ranges (20–45%) with a confident query coverage (highest values varying from 60% to 96%).

Interestingly, all plant protein sequences reported in *Table 1* are related to inflorescence signaling. HVA22 is induced by abscisic acid (ABA)/stress and has a role in the gibberellic acid (GA)-induced cellular death and in the regulation of seed germination (*Shen et al., 2001*). FT is a florigen that induces and promotes the transition from vegetative growth to flowering (*Koornneef et al., 1998*). The protein MFT is involved in regulation of seeds germination by ABA/GA signaling (*Vaistij et al., 2018*). Heading date 3A-like, as FT, is a probable florigen, which promotes the transition from vegetative growth to flowering downstream of HD1 and EHD1 under short day conditions (*Taoka et al., 2011*). It is also remarkable that plant proteins previously reported in the literature (COI1–JAZ, SABP2, and TPL, see Introduction) were not retrieved by our search based on plant–animal protein similarities. This suggests that plant proteins may be able to work as OBPs even if different from animal OBPs, both at primary and tertiary structure levels.

The putative OBPs retrieved by approaches (a) and (b) were added to the three plant OBPs already described in the literature (see Introduction), and all proteins were checked for availability of experimental 3D structure data in the second step of our study. This search was successful for nine

**Table 1.** Similarities of animal and plant protein families with odorant receptor/transporter/channel functions.

| Animal protein family | Similar plant proteins | Identity percentage range | Sequence coverage range | BLAST E-value |
|---|---|---|---|---|
| Bovine cyclic nucleotide gated olfactory channel of *Bos Taurus* | Potassium voltage-gated channel | 25–30% | 60–75% | ≤3e-27 |
| Chloride channel Anoctamin-2 of *Mus musculus* | anoctamin-like protein | 22–27% | 50–70% | ≤3e-20 |
| Human/mouse receptor expression enhancing molecules | HVA22 e HVA22-like plants proteins | 27–44% | 40–96% | ≤7e-21 |
| BPI fold containing family B member three from *Mus musculus* and *Rattus norvegicus* | putative BPI, lipid-binding protein, hypothetical protein, and unnamed protein | 20–30% | 25–60% | ≤6e-07 |
| putative OBP 5a from *Drosophila melanogaster* | FT, D3-like, protein 'Mother of FT and TFL1-like (Terminal flower 1-like)' (MFT), ZCN9 (MFT-like), Heading Date 3A-like | 28–41% | 51–86% | ≤5e-18 |

putative plant OBPs that were then selected for molecular docking simulations of the interactions between potential plant OBPs and selected VOCs, to finally identify candidate plant OBPs.

*Table 2* reports the binding energy values obtained by docking simulations for each complex between potential plant OBPs and ligands (plant VOCs), together with the binding energy values obtained as a reference for experimental complexes after a redocking procedure, when available (see Materials and methods).

The predicted binding constant (Ki) values are reported in *Supplementary file 3* and should be interpreted only as indicative values. Reported Ki values are very high compared to those found for animal OBPs, but are consistent with other plant Ki studies, confirming that plants may sense VOCs only when exposed to higher concentrations than animals (*Nagashima et al., 2019*).

Our results indicate that the three monoterpenes tested (α-pinene, β-myrcene, and limonene) may bind some of the putative OBPs with energy values similar or lower than the values observed for the reference complexes. For example, α-pinene binds the reference protein cytochrome P450 2B6 with an energy value of −5.42 kcal/mol and a predicted Ki of 103 μM. In our docking simulations, α-pinene seems to dock better on SABP2 and on the complete JA receptor, with binding energy value of −6.03 kcal/mol and −5.92 kcal/mol, and predicted Ki of 37 μM and 38.55 μM, respectively. In both cases, this is a better interaction than with the reference complex. In the case of GA receptor and heading date 3A, α-pinene binding energy values were similar to those of the reference complex. Similar to α-pinene, β-myrcene binds SABP2, GA receptor, and the JA receptor

**Table 2.** Binding energy values (Kcal mol$^{-1}$), obtained by docking simulations, between putative plant OBPs and isoprenoid VOCs.

| OBPs | α-Pinene | Limonene | β-Myrcene | β-Caryophyllene | Isoprene | Linalool |
|---|---|---|---|---|---|---|
| TPL-like | −5.06 | −4.76 | −3.95 | −6.16 | −3.39 | −3.94 |
| ABA receptor | −4.69 | −4.73 | −3.84 | −6.32 | −3.35 | −4.12 |
| GA receptor | −5.50 | −5.44 | −4.92 | −6.94 | −3.78 | −4.69 |
| Heading Date 3A | −5.46 | −4.99 | −4.29 | −6.83 | −3.44 | −4.79 |
| FT | −5.23 | −5.07 | −4.26 | −6.65 | −3.15 | −4.98 |
| TFL1 | −5.30 | −5.53 | −4.38 | −7.15 | −3.70 | −4.98 |
| Partial JA receptor | −5.23 | −4.79 | −4.16 | −6.05 | −2.78 | −4.21 |
| Complete JA receptor | −5.92 | −5.11 | −4.41 | −6.61 | −2.82 | −4.49 |
| SABP2 | −6.03 | −6.12 | −5.14 | −6.73 | −3.25 | −4.70 |
| Reference protein* | −5.42 | −6.29 | −4.15 | Not available | Not available | Not available |

*Reference protein is the protein for which it has been found an experimental complex with the ligand. For α-pinene, limonene, and β-myrcene, the reference proteins are cytochrome P450 2B6 complexed with α-pinene (PDB code: 4I91), lipid binding protein complexed with limonene 1,2 epoxide (PDB code: 2A2G, and linalool dehydratase/isomerase complexed with β-myrcene [PDB code: 5HSS]), respectively.

better than the reference complex. Among the other candidate OBPs, protein heading date 3A, FT, and tfl1 showed binding energy values similar to the reference complex for β-myrcene.

The reference complex found for limonene is a modified form of the plant volatile (limonene 1,2 epoxide), which may not interact with the protein-binding site exactly as the VOC does. Therefore, its binding energy value represents a reference point less reliable than the values obtained by the other two experimental protein–ligand complexes. However, as in the previous cases, SABP2 showed energy values lower than the other candidate OBPs, and similar to the reference complex.

Results obtained for β-caryophyllene, isoprene, and linalool cannot be compared to a reference complex, as experimental complexes of these ligands with proteins are not available. In these cases, protein-binding capacity can only be derived by comparing binding values of the different VOCs, and with an even lower confidence. Remarkably, β-caryophyllene showed the lowest, and isoprene the highest, energy binding values.

Our analysis overall confirms that OBPs might be present in plants, and also bind VOCs produced by plants through the MEP pathway. MEP synthesizes isoprenoids that are emitted constitutively (e.g. isoprene) or that are both constitutive and stress induced (e.g. monoterpenes) (*Dicke and Loreto, 2010*). While monoterpenes are efficiently bound by OBPs, isoprene, the simplest and most abundantly emitted volatile isoprenoid, does not seem to bind strongly enough any OBP. Ecological observations report a role for monoterpenes in plant communication with other organisms (*Bouwmeester et al., 2019*), which is arguably not observed for isoprene (e.g. *Brilli et al., 2009*), However, isoprene influences many plant traits (*Monson et al., 2021*) and profoundly modifies properties of cellular and sub-cellular membranes (*Velikova et al., 2015*; *Pollastri et al., 2019*), which may in turn activate signals reshaping plant genomes and phenomes (*Harvey and Sharkey, 2016*; *Miloradovic van Doorn et al., 2020*). As isoprene is the main VOC emitted constitutively and not induced by stresses, it may be tempting to generalize from our observations that, unlike induced VOCs, constitutive VOCs are not bound by OBPs.

Interestingly, monoterpenes seem to bind more efficiently with OBPs that are also reported to bind other plant volatiles. In particular, SABP2, the protein that strongly binds the stress-induced volatile MeSA, also seems to be a candidate for three tested monoterpenes. Protein heading date 3A and tfl1, GA receptor, and FT may also bind, perhaps more specifically, the three monoterpenes. Our results suggest that, as reported for the OBPs from animals and insects (*Ramoni et al., 2007*), the candidate plant OBPs have a broad ligand binding specificity and, consequently, they are likely to bind several different VOCs. This should be tested experimentally by monitoring in vivo the docking patterns of constitutive and induced VOCs.

We noticed that in many cases binding of the ligands occurs at the same protein structure site, as shown for SABP2 in the experimentally reported complex with SA (*Figure 1A*), and in the simulated complexes with α-pinene, limonene, and β-myrcene (*Figure 1B–D*).

The SABP2 binding site (represented in the right panels of *Figure 1B–D*) is characterized by the presence of aromatic side chains (two phenylalanines, one tyrosine, and one tryptophan), also observed in other candidate plant OBPs (GA receptor: two phenylalanines and four tyrosines; JA receptor: one phenylalanine, one tyrosine, and one tryptophan residue; FT receptor: two phenylalanines and one tryptophan). Other candidate plant OBPs have some aromatic side chains in the binding site, although in lower number (e.g., the ABA receptor has one phenylalanine and one tyrosine, while TLF1 has two phenylalanines). This is also reminiscent of the binding site of OBPs from animal organisms (*Bianchet et al., 1996*). Bovine OBPs include five phenylalanines and one tyrosine; *Drosophila* OBPs have four phenylalanines and one tryptophan; and porcine OBPs have two phenylalanines. This conserved feature across biomes may reveal that a hydrophobic environment where odorant molecules can be accommodated is needed. Analysis of the β-caryophyllene complexes (see *Figure 1—figure supplement 1*) also suggests that larger ligands may interact with additional aromatic or hydrophobic side chains in the binding pockets.

Overall, our study confirms that plant OBPs may exist and that they may be structurally and functionally similar to OBPs described in animals. As in the case of animal OBPs, also plant OBPs seem to be able to bind different VOCs in the same binding site, using the same amino acid sequences. While our in silico results make the case that plants also have OBPs, with both common and different features compared to animal OBPs, functional validation should follow. For example, chemical synthesis of fluorescent VOCs could be used to confirm VOC binding by putative OBPs and to characterize protein–ligand binding mechanisms and sites. Mutants or genetically modified plants that miss

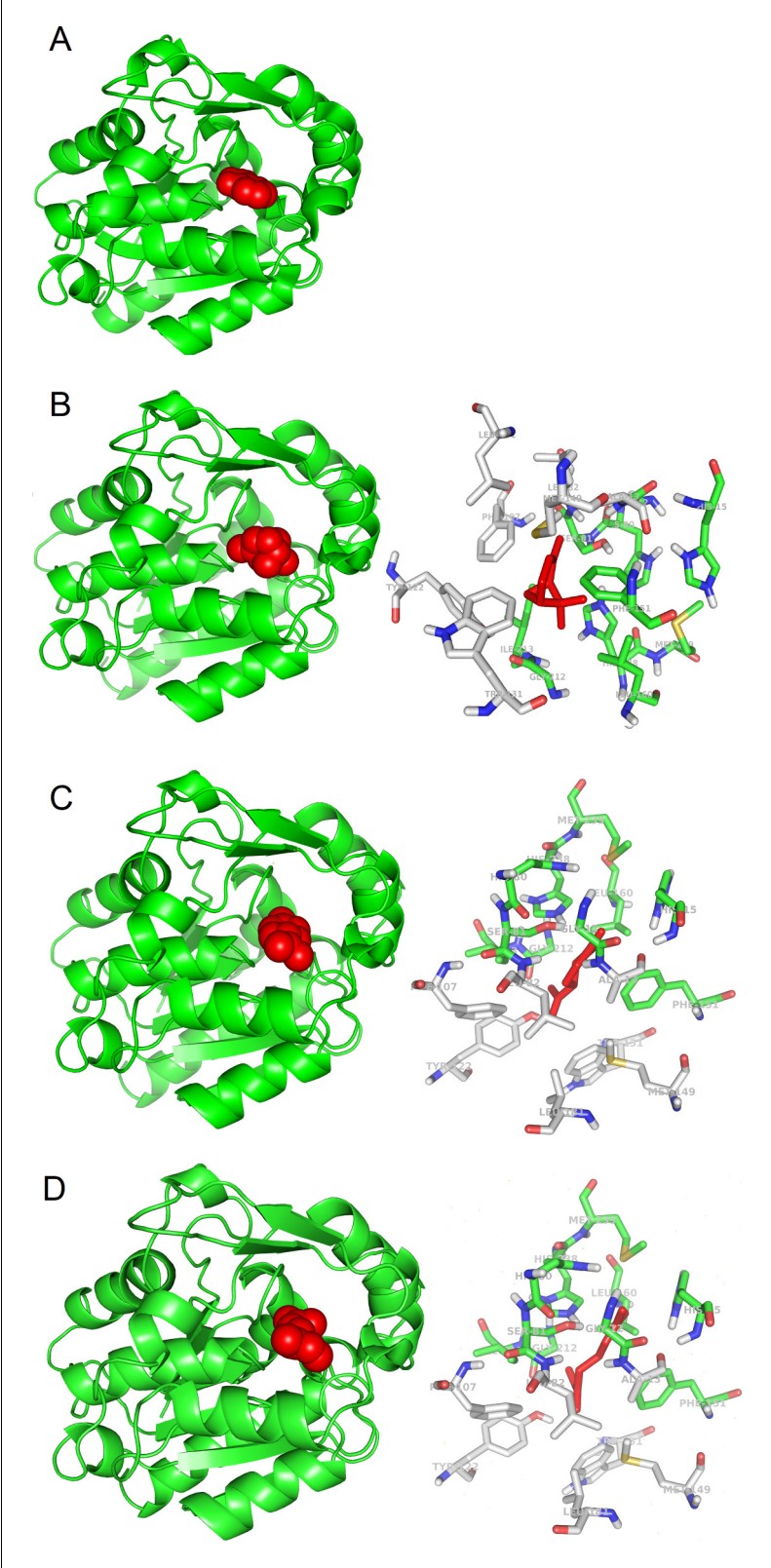

**Figure 1.** 3D models of protein-ligand interactions. (**A**) Experimental structure of SABP2 protein structure with salicylic acid in the binding site (PDB structure 1Y7I). The architecture of SABP2 is schematized by the backbone structure (green), with ribbons and arrows to evidence helices and beta strands, respectively. The salicylic acid molecule (colored in red) is in space-fill representation. (**B**) Left: Molecular docking simulation of SABP2 protein structure with α-pinene (red molecule) in the binding site. SABP2 is shown with the same spatial orientation as in panel (**A**) to emphasize that α-pinene

*Figure 1 continued on next page*

*Figure 1 continued*

occupies the same binding site of salicylic acid. Right: Focus on the binding site of SABP2. α-Pinene (red molecule) interacts directly with amino acid residues labeled with carbon atoms in green. Amino acid residues with carbon atom in grey are also part of the binding site, although not directly in contact with α-pinene. Standard colors are used for the other amino acid atoms (red = oxygen, blue = nitrogen, yellow = sulphur, white = hydrogen). (C) Left: SABP2 protein with limonene (red molecule) in the binding site. SABP2 is shown with the same spatial orientation of (A), to better show that limonene occupies the same binding site of the salicylic acid. Right: Focus of the binding site of SABP2. Details as in panel (B). (D) Left: SABP2 protein with β-myrcene (red molecule) in the binding site. SABP2 is shown with the same spatial orientation of (A), to better show that β-myrcene occupies the same binding site of the salicylic acid. Right: Focus of the binding site of SABP2. Details as in (B).

The online version of this article includes the following figure supplement(s) for figure 1:

**Figure supplement 1.** The environment of the binding pocket of β-caryophyllene in the complexes with best binding energy values.

**Figure supplement 2.** Schematic workflow for the search of candidate odorant-binding proteins in plants.

or have abundant OBP candidates could also be used. Retrieval and description of plant OBPs may be an important step to unveil how plants eavesdrop messages sent by other plants and how the information is then used to activate molecular and metabolic changes leading to defensive responses and patterns.

## Materials and methods

The search for potential OBP proteins in plants was performed following the two-level investigation procedure schematized in *Figure 1—figure supplement 2*. The first level was about searching for plant proteins with potential OBP function. The second level included searching for experimental 3D structures of the candidate plant OBPs and validating by molecular simulations potential ability of putative OBPs to bind isoprenoid VOCs.

In detail, for the first level, two steps were followed. Step (a) was a screening for proteins of interest performed on the UniProt (http://www.uniprot.org) and NCBI (http://www.ncbi.nih.nlm.gov) protein databases. Initial screening was performed by using the protein name and entry annotations, with the query 'odorant-binding protein'. Five plant proteins were found, annotated as 'predicted proteins', which means that they were obtained by nucleotide sequence translation, without evidence at protein or transcript levels, and the name was assigned to the proteins by similarity to other proteins. The protein sequence selected were further investigated by BLAST searches for similar sequences, by using the BLAST interfaces at the database web sites. Standard BLAST search parameters were used. Step (b) was based on BLAST searches (using the same standard parameters of step a) for plant proteins and protein families similar to the 432 OBPs from animal sources available in the protein databases (the list of the 432 OBPs is reported as *Supplementary file 4*).

The second level of investigation was a molecular simulations of the interaction of the potential plant OBPs (i.e. those selected in steps 1a and 1b, and the three proteins for which OBP function has been reported [see Introduction]) with selected isoprenoid VOCs. First, we verified the availability of 3D structures of the candidate OBPs in plants. In particular, the Protein Data Bank (PDB) (http://www.rcsb.org), collecting the 3D structure of proteins, was interrogated for appropriate protein structures of the candidate plant OBPs identified by the first level search. The screening allowed us to select the following plant proteins with potential primary or secondary function as OBP, and with available 3D structures: ABA Receptor from *Arabidopsis thaliana* (PDB code: 4dsb); GA receptor GID1 from *Oryza sativa* (3ebl); Flowering locus t (FT) from *A. thaliana* (1wkp); Terminal flower 1 (tfl1) from *A. thaliana* (1wko); Protein Heading date DATE 3A from *O. sativa* (3axy); TPL-like protein from *A. thaliana* (5nqs); COI (partial JA receptor) from *A. thaliana* (3ogl); COI and JAZ (JA receptor complete) from *A. thaliana* (3ogl); and salicylic acid binding protein 2 (SA enzyme) from *Nicotiana tabacum* (1y7i).

Molecular structures of VOCs were extracted from the PubChem database. The VOCs selected as ligands in our study were the isoprenoids α-pinene (PubChem code: 6654), limonene (22311), β-myrcene (31253), β-caryophyllene (5281515), isoprene (6557), and linalool (6549).

Molecular simulation experiments of protein–ligand interactions were carried out with Autodock 4.2 and AutoDock Tools 1.5.6 (*Morris et al., 2009*), which allowed us to prepare the screening, perform the docking simulation, and analyze the results. Molecular visualization of results was obtained with PyMOL Molecular Graphics System, Version 1.3 Schrödinger, LLC.

The binding energy values obtained for the simulated protein–ligand complexes were compared to the values for complexes used as reference. We found in the PDB database complexes of animal proteins with α-pinene, limonene 1,2 epoxide, and β-myrcene. α-Pinene and β-myrcene are two of the selected VOCs for our simulation, fully correspondent to the natural molecules synthesized and emitted by plants. Limonene 1,2 epoxide is a modified form of the natural VOC. Although not identical to the corresponding plant VOC (limonene), it may be useful as additional reference value. For available plant receptors (ABA receptor, GA receptor, JA receptor, and SA binding protein 2), the reference structures are complexes with ABA, GA, JA-isoleucine, and SA, respectively. These complexes may offer additional reference values of binding energy.

To validate the docking simulation experimental protocol, we applied a redocking procedure to the reference complexes, following the procedure in use in our laboratory (*Scafuri et al., 2016*, *Scafuri et al., 2020*). We depleted the ligand from the complex ligand–protein, and then the ligand-depleted complex (the protein alone) was used to simulate the ligand docking. The redocking experiments were carried out for the protein–ligand reference structures selected above. This approach allowed us to check that the simulation procedure located correctly the ligand in the expected binding site and to calculate the reference value of the binding energy expected in the true protein–ligand complex. The redocking procedure also provided a computational estimation of the binding energy in a true case of protein–ligand interaction. This estimation is used as reference in comparison to the energy binding values computed for the putative protein–ligand interactions. For each ligand a proper reference complex is needed, being the energy of interaction dependent on the ligand chemical features. In the absence of a reference complex relative to an experimental protein–ligand interaction (e.g. the cases of β-caryophyllene, isoprene, and linalool, see *Table 1*), the computed binding energy values may be compared each other too, but only for a qualitative ranking, without a reference threshold given by an effective binding.

## Acknowledgements

FL acknowledges contribution from the Project PRIN – COFIN 2017 (Italian Ministry of University and Research): 'Plant multitROphic interactions for bioinspired Strategies of PEst ConTrol (PROSPECT)'. We would like to thank Prof. Paolo Pelosi for the inspiring discussions about OBPs.

## Additional information

### Funding

| Funder | Grant reference number | Author |
|---|---|---|
| Ministero dell'Istruzione, dell'Università e della Ricerca | PRIN - COFIN 2017 | Francesco Loreto |

The funders had no role in study design, data collection and interpretation, or the decision to submit the work for publication.

### Author contributions

Deborah Giordano, Formal analysis, Validation, Investigation, Visualization, Methodology; Angelo Facchiano, Formal analysis, Validation, Investigation, Methodology; Sabato D'Auria, Conceptualization, Supervision, Writing - original draft, Writing - review and editing; Francesco Loreto, Conceptualization, Supervision, Funding acquisition, Writing - original draft, Writing - review and editing

### Author ORCIDs

Deborah Giordano https://orcid.org/0000-0002-5838-1913
Angelo Facchiano https://orcid.org/0000-0002-7077-4912
Francesco Loreto https://orcid.org/0000-0002-9171-2681

### Decision letter and Author response

Decision letter https://doi.org/10.7554/eLife.66741.sa1
Author response https://doi.org/10.7554/eLife.66741.sa2

# Additional files

## Supplementary files

• Source data 1. Database accession numbers. Related to *Table 1*, *Figure 1* and *Figure 1—figure supplement 1*.

• Source data 2. Docking results. Related to *Figure 1* and *Table 2*.

• Source data 3. PDB structures of SABP2 complexes. Related to *Figure 1*.

• Source data 4. PDB structures of protein - beta caryophyllene complexes. Related to *Figure 1—figure supplement 1*.

• Supplementary file 1. Plant proteins from sequence databases annotated as 'odorant-binding protein'.

• Supplementary file 2. BLAST results for non-plant proteins with similarity to 'general odorant-binding protein 56d' from *Anthurium amnicola*. Only the best ten results are shown.

• Supplementary file 3. Complete results of docking simulation experiments. Docking simulations by AutoDock suite generated results summarized in the table. For each receptor-ligand simulation, the best 100 conformations are clusterized by AutoDock on the basis of ligand position, and three values are reported for each cluster: the mean binding energy for the clusterized conformations, the best binding energy, the number of conformations in the cluster. An additional parameter computed by AutoDock is the predicted $K_i$ value. We reported the results for the best cluster in terms of energy and population, or in some cases, alternative clusters, with reference to different pockets on the receptor surface.

• Supplementary file 4. List of 432 protein sequences selected as OBP from animal sources, used for searching public database for similar plant proteins.

• Transparent reporting form

## Data availability

All data generated during this study were obtained by analysing entries retrieved from public databases, according to procedures described in the manuscript. Source data and supporting files report complete list of accession numbers of entries and the models of 3D structures obtained by our study and used for generating figures.

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
