## [Decision Letter]

**Acceptance summary:**

The article framed as a hypothesis is much clearer than the previous version. The concept of plant OBPs and the potential role in chemical signalling in plants may stimulate further research in this area. The molecular modelling gives intriguing hints that will need to be verified experimentally.

**Decision letter after peer review:**

[Editors’ note: the authors submitted for reconsideration following the decision after peer review. What follows is the decision letter after the first round of review.]

Thank you for submitting your work entitled "On the capacity of putative plant odorant-binding proteins to bind volatile plant isoprenoids" for consideration by *eLife*. Your article has been reviewed by 3 peer reviewers, one of whom is a member of our Board of Reviewing Editors, and the evaluation has been overseen by a Senior Editor. The reviewers have opted to remain anonymous.

We are sorry to say that, after consultation with the reviewers, we have decided that your work will not be considered further for publication by *eLife*. We do think that the topic is exciting and timely. We agree that the proposed ideas are plausible and that the in silico analysis is a helpful step towards investigating these ideas. However, we have come to the conclusion that the in silico analysis is not, by itself, sufficient for a short report in *eLife*, because it leaves too many open questions to either cleanly reject, or substantially develop the authors' hypotheses. The hypotheses themselves are well chosen but not highly novel, and so such a "hypothesis curation" function would be especially important to make the paper of great value for the community.

Summary:

The chemical sensing mechanisms of plants, which are largely unknown, are a topic of broad interest. The authors hypothesise that plant chemical receptors may be transporter proteins or odorant binding proteins analogous to those found in animals. The authors have identified a list of plant proteins with possible odorant binding activity and they predict binding constants for relevant odorants. The calculated binding constants are generally very weak in comparison to known animal odorant binding proteins (i.e., would require much higher concentrations of odor for detection). The in silico investigation, while inspiring, leaves many open questions, for example whether or not there is evidence for functional analogy between plant and animal odorant binding proteins.

*Reviewer #1:*

The chemical sensing mechanisms of plants are largely unknown. The authors hypothesise that plant chemical receptors may be transporter proteins or odorant binding proteins. The authors carried out an "in silico" analysis to investigate whether there are analogous proteins in plants to odorant binding proteins found in animals and insects. A search through protein databases found 5 possible sequences or partial sequences and these were used for further screening using BLAST software for screening comparison. The search for OBPs in plants based on literature evidence and sequence similarities to known OBPs from animal organisms or database annotations, produced a list of plant proteins, or protein families, with potential OBP activity. Molecular docking simulation experiments, to identify candidate plant OBPs were carried out identifying the ligand binding sites and calculated binding energies together with binding constants. This data mining activity has produced some interesting data but also raises several questions. The majority of binding constants tabulated were in the hundreds of micromolar to millimolar concentrations which raises the question of what concentrations of volatile chemicals are plants able to detect. Most odorant binding proteins found in insects and animals have binding constants in the low micromolar range for the target analytes. So if these putative plant odorant binding proteins do have a role in chemical sensing further practical experiments are needed and these need to be discussed. The data are not enough to indicate that these putative proteins identified have relevant functions in chemical sensing in plants.

The manuscript is quite interesting but lacks detail and adequate discussion of the results.

1. Binding pocket characterisation, shape, size – dimensions of pockets and mouth, which amino acids are likely to interact with the ligand, would give some idea as to how the ligands selected fit into the pockets found.

2. Predicted binding constants indicate in general sensitivity in the high micromolar to mm ranges of ligands. This has implications for how a plant is able to respond to low concentrations of volatiles that are emitted from other plants – some discussion is needed.

3. β-caryophyllene may have better affinity of binding than other compounds – it may be interesting to examine more carefully the fit of this molecule in the binding pockets and which amino acids are involved.

3. The data presented do not give evidence for the presence of plant OBPs as claimed, as the proteins involved may be involved in other plant signalling functions. In presenting the data it would not be wise to make over ambitious claims as experimental data to validate the potential OBPs is missing.

4. The Discussion should have a section on what experiments are needed to test and validate the data presented.

5. There are a few typographical and grammatical errors that should be corrected. eg β-caryophillene should be β-caryophyllene, salicylic acid sometimes is spelt incorrectly etc.

*Reviewer #2:*

The Authors started from the consideration that volatiles emitted by plants may serve as communication media to other plants. Hence, the 'receiver' plant needs a way to bind these molecules and initiate the transduction cascade, that is a dedicated protein that is able to bind volatiles because it has a binding groove able to accommodate the ligand. The Authors searched the available databases for putative proteins that can serve this goal, by similarity to already known proteins from plants and animals. Then, by molecular docking known plant volatiles, the Authors demonstrate that the identified proteins have the predicted structural features that allow ligand binding.

The main strength of the paper resides in the idea and in the wide search for similarities among proteins pertaining to different kingdoms. The main weakness of this work resides in the fact that it is entirely in silico, without providing any actual data from real proteins. However, it is well known that in silico simulations may be only suggestive of the actual behavior of a protein.

The Authors reached their goal of identifying various proteins with putative binding abilities, however the lack of any experimental data should be made clear since the beginning of the manuscript, in the title and abstract. With this caveat, the information provided in this paper may be a useful starting point for experimentally testing of the hypotheses.

In addition to the above mentioned comments, I underline that in some instances the terminology used by the Authors should be more accurate and revised according to the following suggestions:

– Title: made clear that this paper contains only in silico simulations.

– Line 28: 'talk' refers to a specific form of communication involving voice. Substitute with 'communicate' or a synonym.

– Line 1: 'perceive' means being conscious of the presence of a sensory stimulus. Unse instead 'sense', 'detect' or a synonym.

– Line 33: add 'plant' between 'whether' and 'OBP'.

– Line 57: use 'detect' instead of 'perceive'.

– Line 87: what is 'capacity'?

– Table 1: I am not aware of any 'Bovine Cycline Nucleotide Gating Olfactory channel'. Maybe the authors refer to: 'Bovine Cyclic Nucleotide Gated Olfactory channel'?

– Table: what is the 'similarity ' reported here? Please add the level (percentage?) of similarity or identity of the proteins.

– Supplementary table S3: in the 'predicted Ki' column, the numbers are separated by a comma. Should it be full stops instead?

An additional more general comment refers to the lack of information on Odorant binding proteins. In particular, since they refer similarity to animal OBP, the authors should made clear that animal OBP are very different and pertain to different classes with dissimilar sequence and 3D structure. A couple of sentences and reference to a review on this topic may help in setting the stage.

*Reviewer #3:*

It is known that stress-induced plant volatiles can be perceived by neighboring plants, but the underlying mechanism is largely unknown. The authors of this manuscript attempted to identify receptors that interact with isoprenoids, the most abundant plant stress-induced volatile organic compounds (VOCs). They established a framework that allowed them to screen for plant odorant-binding proteins (OBPs) through all available databases. Comparing plant protein sequences with a large group of animal OBP sequences and expanding the investigation to previously known OBPs turned out to be fruitful. Molecular simulation is a powerful screening technique to study the interaction of the potential plant OBPs with selected isoprenoids. The finding that plant OBPs may bind different VOCs in the same binding site is interesting.

The in silico selection of the plant OBP candidates and ligand docking experiments provide a useful tool to understand signals that underpin plant-plant interactions as well as how plants respond to those cues. However, the conclusions made by the authors may be too premature:

Isoprenoids are both constitutive and stress-induced. The authors did not address, through such a docking study, how plants distinguish VOCs associated with impending stresses, particularly when the OBPs could generally interact with multiple VOCs. One may also wonder how many other types of VOCs exist that are stress-responsive, and what their receptors are.

The BLAST of 432 OBPs did not find the JA receptor, suggesting that the JA receptor sequence is not closely related to that of the OBPs that insects use to recognize plant volatiles in order to locate suitable host plants. Therefore, identification of OBPs based on sequence similarity may miss those plant proteins that possess OBP structure and function but differ in primary sequences with existing OBPs.

Docking studies can serve as a lead for potential OBP-VOC interaction, but in silico data alone is insufficient to conclude the role of the putative OBPs. Functional evidence is necessary to demonstrate their interaction with VOCs because such interaction indeed affects plant response.

Table 2 revealed the binding energy values between the putative plant OBPs and isoprenoid VOCs. Since OBPs were not selected based on specificity for isoprenoid VOCs, will it be worthwhile testing other VOC classes?

Functional validation following the lead from the in silico study should be included, e.g. via mutant or overexpression studies.

---

## [Author Response]

[Editors’ note: The authors appealed the original decision. What follows is the authors’ response to the first round of review.]

In this version we have taken into consideration constructive comments and criticisms of editors and reviewers. In details:

The editors asked us that, as our in silico study can suggest but cannot test that OBPs are present in plants and bind VOCs, we frame a “hypothesis paper”, moderating our claims, cautioning the readers about limitations of our approach and recommending future investigations to demonstrate our hypothesis.

We thank the editors for their very wise suggestions. We have amended the revised text accordingly. Starting with the title and throughout the paper, we have stated now that our study only offers a novel hypothesis. Moreover, we have highlighted research that must be accomplished to test whether our hypothesis holds true in nature (e.g. line 179 and 219, also responding to similar suggestions by Ref. 2 and 3). We have also added a further in silico test, supporting our argument that binding pockets of hydrophobic protein side chains may accommodate even larger ligands, e.g. volatile sesquiterpenes (Figure 1—figure supplement 1).

The editors asked us to keep the “short report format” for the paper. To meet this request, we have largely reshaped the paper, moving technical information to the methods section or to supplementary materials. Table 1 was enriched with details requested by the reviewers (see below) and Figure 1 was redrafted showing important and conserved details of molecular docking simulations between VOCs and the SABP2 putative OBP.

The editors asked us to consider those useful comments of the reviewers, other than missing experimental data. We have carefully followed this recommendation. In particular:

Ref. 2 wrote that “the lack of any experimental data should be made clear since the beginning of the manuscript, in the title and abstract”.

As mentioned above, we have reshaped the text in a more conservative way, starting from title, which now opens with the word “hypothesis” and abstract, where a sentence was added to clearly state that our work is based on molecular docking tests and that OBPs identified are only putative until experimentally tested.

Ref.3 commented: “The BLAST of 432 OBPs did not find the JA receptor, suggesting that the JA receptor sequence is not closely related to that of the OBPs that insects use to recognize plant volatiles in order to locate suitable host plants. Therefore, identification of OBPs based on sequence similarity may miss those plant proteins that possess OBP structure and function but differ in primary sequences with existing OBPs.”

We thank the reviewer for this insightful observation. What the reviewer says is also true for other putative plant OBPs and we added sentences to suggest that plant proteins may be able to work as OBPs even if different from animal OBPs, both at primary and tertiary structure level (line 117).

Ref. 2 commented about Table 1: “what is the 'similarity ' reported here? Please add the level (percentage?) of similarity or identity of the proteins”.

To answer to this valuable request, three columns were added to Table 1, showing the identity coverage range, the sequence coverage range and the BLAST E-value. In the text (line 104), it is also stated how sequence similarity was assessed based on these indicators.

Ref. 1 felt that: “Binding pocket characterisation, shape, size – dimensions of pockets and mouth, which amino acids are likely to interact with the ligand, would give some idea as to how the ligands selected fit into the pockets found”.

Ref. 1 also noted that “β-caryophyllene may have better affinity of binding than other compounds – it may be interesting to examine more carefully the fit of this molecule in the binding pockets and which amino acids are involved.”

These are difficult requests, as volume and size of the pockets may vary considerably depending on the used software. However, prompted by the reviewer’s observations, we have expanded the part dealing with binding site characterization. We already described in our manuscript the presence of aromatic side chains in the binding pockets, and now we added a specific focus on the sesquiterpene β-caryophyllene (Figure 1—figure supplement 1 and line 194) to show that even large ligands may theoretically bind aromatic or hydrophobic side chains in the binding pockets of putative OBPs.

Ref.1 wrote that: “The majority of binding constants tabulated were in the hundreds of micromolar to millimolar concentrations which raises the question of what concentrations of volatile chemicals are plants able to detect. Most odorant binding proteins found in insects and animals have binding constants in the low micromolar range for the target analytes. So if these putative plant odorant binding proteins do have a role in chemical sensing further practical experiments are needed and these need to be discussed.”

Note also ref. 2 suggestion: “An additional more general comment refers to the lack of information on Odorant binding proteins. In particular, since they refer similarity to animal OBP, the authors should made clear that animal OBP are very different and pertain to different classes with dissimilar sequence and 3D structure. A couple of sentences and reference to a review on this topic may help in setting the stage.”

We thank the reviewers for asking us to focus on possible differences between plant and animal OBPs. We make clear now that binding constant should be treated very cautiously (and this is why they are presented as Supplementary Material). But we also acknowledge that our Ki values are higher than in animal OBPs. Yet, our values are similar to those reported in the few other studies about plant OBPs. This raises the interesting question whether plants sense VOCs only when exposed to higher concentrations than animals. We discuss this in the revised text (line 135).

Ref. 3 asked: “Table 2 revealed the binding energy values between the putative plant OBPs and isoprenoid VOCs. Since OBPs were not selected based on specificity for isoprenoid VOCs, will it be worthwhile testing other VOC classes?”

This is a valid comment, and we could clearly try. However, we need to have the crystallographic structures of a protein complexed with the selected VOC, to be able to calculate reference binding energy values by redocking experiments. Moreover, we wanted to keep the paper focused on volatile isoprenoids for their very large biological and environmental interest and for being molecules largely emitted by plants both constitutively and in an induced manner. Enlarging the investigation to more VOCs will surely be a future endeavor.

Ref. 3 noted that “Isoprenoids are both constitutive and stress-induced. The authors did not address, through such a docking study, how plants distinguish VOCs associated with impending stresses, particularly when the OBPs could generally interact with multiple VOCs. One may also wonder how many other types of VOCs exist that are stress-responsive, and what their receptors are.”

We actually think that this is a main finding of the paper. Among the studied VOCs, isoprene is constitutive, and the other isoprenoids are mainly induced. We argue that isoprene and perhaps other constitutive VOCs that are emitted life-long by plants do not have sufficient binding properties, contrary to induced VOCs. This was clearly stated in the previous version (e.g. Abstract, line 37). However, to strengthen further this important idea behind our results, we entered a new sentence in the discussion (line 170).